# Effect of Orthopedic Treatment for Class III Malocclusion on Upper Airways: A Systematic Review and Meta-Analysis

**DOI:** 10.3390/jcm9093015

**Published:** 2020-09-18

**Authors:** Golnar Havakeshian, Vasiliki Koretsi, Theodore Eliades, Spyridon N. Papageorgiou

**Affiliations:** Clinic of Orthodontics and Pediatric Dentistry, Center of Dental Medicine, University of Zurich, 8032 Zurich, Switzerland; g.havakeshian@hotmail.com (G.H.); koretsi.vasiliki@gmail.com (V.K.); theodore.eliades@zzm.uzh.ch (T.E.)

**Keywords:** class III malocclusion, maxillary retrognathism, orthopedic treatment, dentofacial orthopedics, orthodontics, clinical trials, systematic review, meta-analysis

## Abstract

The aim of this systematic review is to compare the effect on the upper airways of orthopedic treatment for skeletal Class III malocclusion with untreated controls. Nine databases were searched up to August 2020 for randomized or nonrandomized clinical trials comparing orthopedic Class III treatment (facemask or chin-cup) to untreated Class III patients. After duplicate study selection, data extraction, and risk of bias assessment (Risk Of Bias In Non-randomized Studies-of Interventions [ROBINS-I]), random-effects meta-analyses of Mean Differences (MDs)/Standardized Mean Differences (SMD) and 95% Confidence Intervals (CIs) were performed, followed by the Grading of Recommendations Assessment, Development and Evaluation assessment evidence-quality. A total of 10 papers (9 unique nonrandomized studies) with 466 patients (42.7% male; average age 9.1 years) were finally included. Limited evidence indicated that compared to normal growth, maxillary protraction with facemask was associated with increases in total airway area (*n* = 1; MD = 222.9 mm^2^; 95% CI = 14.0–431.7 mm^2^), total nasopharyngeal area (*n* = 4; SMD = 1.6; 95% CI = 1.2–2.0), and individual airway dimensions (upper-airway MD = 2.5 mm; lower-airway MD = 2.1 mm; upper-pharynx MD = 1.6 mm; lower-pharynx MD = 1.0 mm; all *n* = 6). Subgroup/meta-regression analyses did not find any significant effect-modifiers, while the results were retained 2–5 years postretention. Our confidence in these estimates was, however, very low, due to the inclusion of nonrandomized studies with methodological issues. Limited data from 2 chin-cup studies indicated smaller benefits on airway dimensions. Existing evidence from controlled clinical studies on humans indicates that maxillary protraction for skeletal Class III treatment might be associated with increased airway dimensions, which are, however, mostly minor in magnitude.

## 1. Introduction

### 1.1. Background

Skeletal Class III is multifactorial entity consisting of maxillary retroganthism, mandibular prognathism, or a combination thereof. Its treatment often poses a challenge for the orthodontist. Severe skeletal Class III might considerably affect many aspects of the patient’s life, including, among others, psychosocial status, esthetics, mastication, speech and even breathing. The latter might be associated with significant differences in the morphological characteristics of the oropharyngeal airway of patients with Class III deformity. Data from Cone Beam Computerized Tomography (CBCT) indicate that Class III adults might have greater total oropharyngeal volume, but also greater constriction areas in the retroglossal and retropalatal compartment of the oropharynx [1]. The most constricted area of the oropharyngeal airway is usually located at the base of the tongue [2]. At the same time, a positive correlation has been reported among tongue volume, pharyngeal airway and Sella-Nasion-B point (SNB) angle, while a negative correlation has been reported between tongue volume and A point-Nasion-B point (ANB) angle [3]. Finally, Class III patients might have higher risk of being mouth breathers than Class I patients (32% and 23%, respectively) [4].

From the available Class III treatment options, orthopedic modification as well as orthognathic surgery might be expected to exert the greatest influence on the anatomy and volume of the airway. Specifically, significantly more orthognathic patients report a perceived improvement in breathing through maxillary advancement (with or without mandibular setback) (95%) than with mandibular setback alone (56%) [5]. Bimaxillary surgery results in greater increases than mandibular setback in the volume of the airway, which are primarily apparent due to an increase in the retropalatal region [6]. For growing Class III patients, orthopedic appliances are sometimes used to protract the maxilla, restrict or redirect mandibular growth, or both, while at the same time, both approaches lead to a mandibular rotation. However, consensus regarding the impact that these therapeutic approaches might have on the airway has not been reached in the literature. Some studies have reported a favorable effect on the dimensions of the airway [7,8,9,10], while others have not confirmed such an effect [11,12]. A recent systematic review on the topic [13] reported that maxillary protraction can increase postpalatal and nasopharyngeal airway dimensions in growing skeletal class III subjects with maxillary retrusion. However, that review included studies published only up to 2017, and its conclusions might be influenced by many existing issues like the lack of an a priori protocol [14], incomplete handling of risk of bias within studies according to the latest Cochrane guidelines [15], issues with the data synthesis (improper model selection, double-counting of multiple arms from single studies and a lack of sensitivity analyses) [16], and the lack of a formal assessment of the review’s quality of evidence [17]. Finally, that review only assessed the impact of maxillary protraction on airway dimensions and did not include mandibular restriction with chin-cup.

### 1.2. Objective

The aim of this systematic review was to assess the effect of different orthopedic appliances on the upper airway dimension in growing patients with class III malocclusion. The research question was “can orthopedic treatment of growing Class III patients with facemask or chin-cup modify the upper airway dimensions to a degree greater than what can be attributed to normal growth?”.

## 2. Materials and Methods

### 2.1. Protocol and Registration

A review protocol was designed and registered a priori (PROSPERO CRD42020148142), while post hoc changes were transparently reported (Appendix A). The conduct and reporting of this review is based on the Cochrane Handbook [18] and PRISMA statement [19], respectively.

### 2.2. Eligibility Criteria

Clinical studies on human pre-adolescent and adolescent patients of any sex or ethnicity with skeletal Class III malocclusion as an indication for orthopedic treatment were included (Appendix A). No limitations concerning language, publication year, or status were applied. The primary outcome of this review was total dimensions of the upper airways. Secondary outcomes included other dimensions of the separate airway compartments in linear or areal measurements.

### 2.3. Information Sources and Search

Nine electronic databases were systematically searched without any restrictions for publication date, type, and language from inception up to 4 August 2020 (Appendix A), while Directory of Open Access Journals, Digital Dissertations, metaRegister of Controlled Trials, World Health Organization, and Google Scholar, as well as the reference lists of eligible articles or existing systematic reviews were manually searched for any additions.

### 2.4. Study Selection

Two authors (G.H., V.K.) screened the titles and/or abstracts of studies retrieved from the searches to identify articles that potentially met the inclusion criteria, before moving to their full-texts. Any differences between the two reviewers were resolved by discussion with a third author (S.N.P.).

### 2.5. Data Collection Process and Items

Data collection from the identified reports was conducted using predefined and piloted forms covering: (a) study characteristics (design, clinical setting, country), (b) patient characteristics (age, sex), (c) inclusion criteria/malocclusion characteristics, (d) treatment characteristics (appliance and duration), (e) post-treatment follow-up duration, and (f) outcome assessment. Data were extracted by two authors (G.H., V.K.) with the aforementioned way to resolve discrepancies.

### 2.6. Risk of Bias of Individual Studies

The risk of bias of included nonrandomized studies was assessed with the ROBINS-I (“Risk Of Bias In Nonrandomized Studies—of Interventions”) [15]. Assessment of the risk of bias within individual studies was likewise independently performed by two authors (G.H., V.K.) with the same approach being applied to resolve discrepancies.

### 2.7. Data Synthesis and Summary Measures

An effort was made to include all existing trials in the analysis; where data were missing, they were calculated by us (Appendix A). As duration of orthopedic effects on the airways might be affected by patient anatomy or growth potential, appliance characteristics and patient compliance, a random-effects model was deemed appropriate to calculate the average distribution of true effects [20] with a restricted maximum likelihood variance estimator [21], using Mean differences (MDs) or Standardized Mean Differences (SMDs) and their corresponding 95% confidence intervals (CIs). The produced forest plots were augmented with contours denoting the magnitude of the observed effects to assess heterogeneity, clinical relevance and imprecision [16] (Appendix A).

The extent and impact of between-study heterogeneity was gauged visually and quantified with tau^2^ (absolute heterogeneity) and the I^2^ statistic (relative heterogeneity; inconsistency). Inconsistency over 75% was arbitrarily considered as high, but we also considered where the inconsistency was localized on the forest plot and our uncertainty around these estimates [22]. Ninety-five per cent predictive intervals, integral in the correct interpretation of random-effects meta-analyses [23], were estimated for meta-analyses of ≥3 trials to incorporate observed heterogeneity and provide a range of possible effects for a future clinical setting.

### 2.8. Additional Analyses and Risk of Bias across Studies

Many subgroup and meta-regression analyses were originally planned (Appendix A), but only some could be performed in the end. Likewise, reporting biases were planned but could not be assessed in this review.

The overall quality of meta-evidence (i.e., the strength of clinical recommendations) was rated using the Grades of Recommendations, Assessment, Development, and Evaluation (GRADE) approach [17] following recent guidance on synthesizing nonrandomized studies [24] and summary of findings tables were constructed using an improved format [25] (Appendix A).

Robustness of the results was planned to be checked a priori with sensitivity analyses based on (a) baseline similarity of treated-control group in airway measurements, and (b) sample size, while some sensitivity analyses could not be performed (Appendix A).

All the analyses were run in Stata version 14.0 (StataCorp LP, College Station, TX, USA) by one author (SNP) with an openly provided dataset [26]. All *p* values were two-sided with α = 5%, except for the test of between-studies or between-subgroups heterogeneity, where α-value was set at 10% [27].

## 3. Results

### 3.1. Study Selection

The electronic literature search yielded 849 results, while one study was manually identified (Figure 1). After duplicate removal and screening of titles/abstracts against the predefined eligibility criteria (Appendix A), the full texts of 104 papers were checked. Eventually, 10 papers pertaining to 9 unique studies (1 prospective and 9 retrospective nonrandomized studies; Table 1), which were published as journal papers, were finally included [8,11,12,28,29,30,31,32,33,34].

### 3.2. Study Characteristics

All primary studies were conducted in university clinics (*n* = 9; 100%) and originated from four different countries (Iran, Italy, Spain, Turkey) (Table 1). A total of 321 treated and 145 untreated Class III patients were included, with a median total sample of 45 patients per included study (range 34 to 78 patients per study). All studies reported patient sex and age, with 199 of 466 patients (42.7%) being male and the mean patient age being 9.1 years.

Seven of the included studies assessed maxillary protraction with facemask/reverse headgear, one assessed mandibular restraint with chin-cup, and one included one facemask arm and one chin-cup arm. Among the eight facemask studies, four incorporated maxillary expansion, two included a bite-plane (or bite-blocks) and one included both maxillary expansion and bite-plane. Average treatment duration was 15.1 months for facemask treatment (7 studies; range 6.9 to 24.0 months) and 9.8 months for chin-cup (1 study), while the average observation period for the untreated controls was 14.7 months (9 studies; 6.0 to 25.2 months). All studies measured outcomes before and directly after treatment, while two studies additional remeasured outcomes postretention after an additional average period of 2.1 to 5.1 years. Finally, all studies assessed the effects of treatment on airways with lateral cephalograms.

### 3.3. Risk of Bias within Studies

The included nonrandomized trials presented several issues influencing their risk for bias (Table 2). All studies except one (*n* = 8; 89%) were retrospective, and patient selection could influence the results of treatment for most of them (*n* = 8; 89%). For at least three studies (33%), treated/control patients were followed/observed for different durations, while baseline differences in age, sex, malocclusion severity or airway measurements existed for 1–3 studies (11–33%). No study performed blinded outcome measurement, while for at least four studies (44%), the treated and control populations originated from different sources. All included studies were judged to be in critical risk of bias, as issues existed for at least three domains per study.

### 3.4. Results of Individual Studies and Data Synthesis

#### 3.4.1. Maxillary Protraction with Facemask/Reverse Headgear

Eight studies provided various measurements of linear distances or area measurements of airways post-treatment or postretention. The results of the performed meta-analyses are given in Table 3, while the results of single studies that could not be pooled in meta-analyses are given in Appendix A.

The review’s primary outcome of total airway area was assessed by a single study, which found a statistically and clinically relevant increase in airway area post-treatment (MD = 222.86 mm^2^; 95% CI = 14.04–431.68 mm^2^; *p* = 0.04). As far as secondary outcomes, statistically significant post-treatment increases were seen for the facemask group compared to the control group regarding total nasopharyngeal area (4 studies; SMD = 1.62; 95% CI = 1.20–2.04; *p* < 0.001; I^2^ = 23%; Appendix A), aerial nasopharyngeal area (2 studies; MD = 1.29 mm^2^; 95% CI = 0.80–1.77 mm^2^; *p* < 0.001; I^2^ = 0%), upper airway dimensions (6 studies; MD = 2.45 mm; 95% CI = 0.97–3.92 mm; *p* = 0.001; I^2^ = 87%; Appendix A), lower airway dimensions (6 studies; MD = 2.10 mm; 95% CI = 1.50–2.70 mm; *p* < 0.001; I^2^ = 5%; Appendix A), McNamara’s upper pharynx dimensions (6 studies; MD = 1.59 mm; 95% CI = 0.57–2.62 mm; *p* = 0.002; I^2^ = 73%; Appendix A), and McNamara’s lower pharynx dimensions (6 studies; MD = 1.02 mm; 95% CI = 0.17–1.88 mm; *p* = 0.02; I^2^ = 70%; Appendix A). Observed heterogeneity was acceptable in all instances, except for the meta-analysis of upper airway dimensions (I^2^ = 87%), but all studies were on the right side of the forest plot (Appendix A), so judgment about the beneficial effects of treatment was not influenced by it, but rather, only the precise quantification of the treatment’s effects. The 95% prediction intervals were inconsistent (included both negative and negative values) for most meta-analyses, with only total pharyngeal area and lower airway dimension being consistent, meaning that we can consistently expect significant benefits in every future scenario.

Apart from these meta-analyses, individual studies reported on several increases in airway dimensions (Appendix A) that were either statistically nonsignificant or clinically nonrelevant, apart from a relevant increase in oropharynx dimensions.

Finally, postretention data 2.1 to 5.4 years after treatment were available from two studies (Table 3). Lower adenoid side was significantly lower in treated patients compared to controls (2 studies; MD = −2.67 mm; 95% CI = −4.63–−0.70 mm; *p* = 0.008), a novel finding that was not seen directly post-treatment. The dimensions of the upper and lower airways were still significantly larger among treated than untreated patients and to a greater extent than directly post-treatment (MD of 3.71 mm versus 2.45 mm for the upper airway; MD of 3.59 mm versus 2.10 mm for the lower airway). Treatment-related benefits in the dimensions of the pharynx were likewise increased postretention, though this increase was statistically significant only for the upper pharynx.

#### 3.4.2. Mandibular Restraint with Chin-Cup

Two studies provided various measurement on linear distances or area measurements of airways post-treatment (Appendix A), most of which were not significant—with two exceptions. One study reported a post-treatment increase of the nasopharyngeal area (MD = 10,183.0 mm^2^; 95% CI = 10,074.3–10,291.8 mm^2^; *p* < 0.001), which was both statistically significant and clinically relevant. The same study reported a post-treatment reduction of the oropharyngeal area (MD = −8231.0 mm^2^; 95% CI = −10,616.5–−5845.5 mm^2^; *p* < 0.001), which was statistically significant but clinically irrelevant.

### 3.5. Additional Analyses, Risk of Bias across Studies, and Quality of Evidence

Several subgroup analyses, meta-regressions and assessments for reporting biases were originally planned in the protocol, but could ultimately not be performed (Appendix A). Selected subgroup and meta-regression analyses on the post-treatment effects of maxillary protraction (for meta-analyses with ≥5 studies) found no significant influence of patient age, sex, baseline airway dimensions, inclusion of maxillary expansion or treatment duration (Table 4). Likewise, one study found no significant differences between normodivergent or hyperdivergent patients, and another study found no significant differences between two facemask designs.

The quality of evidence (Table 5) for the main analyses on the post-treatment effects of facemask was, in all cases, very low, due to the inclusion of retrospective nonrandomized studies with critical risk of bias. The quality of evidence about the primary outcome of total airway area was additionally downgraded due to imprecision, as a single study with a limited sample size contributed to this. The GRADE analysis indicates that further research in terms of well-designed studies is very likely to have an important impact which will likely change our current estimates of effect.

### 3.6. Sensitivity Analysis

A sensitivity analysis according to sample size and baseline similarity of treated/control patients (Appendix A) indicated relative robustness of the results.

## 4. Discussion

### 4.1. Evidence in Context

The current review summarizes and critically appraises existing evidence from clinical research comparing the effects of Class III orthopedic treatment on airway dimensions to untreated Class III controls. A total of 10 papers (9 studies) including 321 treated and 145 untreated Class III patients were finally identified as eligible and contributed to data synthesis.

Maxillary protraction with a facemask, with or without an expander, was shown to result in statistically significant increases in airway dimensions directly after treatment compared to what could be expected by Class III growth alone. Specifically, benefits were seen for total nasopharyngeal area, upper/lower airway dimensions and upper/lower pharynx dimensions (Table 3). However, most of these changes were small to moderate in magnitude, which means that they might have little clinical relevance (Appendix A). The only exception was the increase in total nasopharyngeal area, where a large to very large effect was (Appendix A) found. This indicates that any clinically relevant benefits in airway dimensions or breathing might be located in this compartment. Evidence from the existing literature indicates that significant differences exist in the dimensions of the pharyngeal airway and the thickness of the pharyngeal wall among normal patients and patients with sleep-disordered breathing [35], with the lower retropalatal and retroglossal areas being affected the most [36]. Even though these regions were only minimally affected, the large improvements at the nasopharyngeal area might be highly relevant, since the nasopharyngeal dimensions appear to be the most sensitive parameter for assessments of the patient’s respiratory conditions [37]. However, even though dimensional changes might be indicative of improved breathing, proper confirmation must follow using functional analyses of nasal airflow resistance and nasal pressure.

The initial effects on the airways observed after treatment were, for the most part, retained after a follow-up period of 2–5 years, while in some instances, the difference between treated and untreated Class III patients even increased (Table 3). This might indicate that early orthopedic modification of a maxillary retrusion might be associated with a more favorable growth pattern, even though this is speculative at the present time. A long-term follow-up from a well-known randomized trial on maxillary protraction indicated relatively similar change patterns for SNA, SNB and ANB angles from 8 to 14 years of age for treated and untreated Class III patients, with mandibular rotation being the most pronounced difference [38]. Different extents of mandibular rotation, as expressed through the patient’s growth pattern, were observed to be differently related to airway dimensions in Class I patients [39]. Apart from that, therapeutic effects should always be considered in the context of physiological growth. The dimensions of the nasopharyngeal compartment are not stable during growth [40,41], and patient age at the time of the investigation could play a role in the interpretation of the results.

The effects of orthopedic maxillary traction on the airways were relatively consistent, despite the big variability of the included studies regarding patient characteristics (age, sex, baseline airway dimensions) and treatment protocol (simultaneous use of maxillary expansion or treatment duration), as no clear modification was seen through the subgroup/meta-regression analyses (Table 4). Previous studies have indicated that maxillary expansion can have a beneficial effect on upper airway dimensions [42], but whether additional gains can be expected during maxillary protraction by also incorporating expansion remains questionable. It must be also noted here that alternating rapid maxillary expansion and constriction might be more beneficial than conventional rapid maxillary expansion in terms of skeletal effects [43], but no included studies used this protocol.

Restricting or redirecting mandibular growth with a chin-cup had a considerably smaller impact on the airways than maxillary protraction, since only two statistically significant differences directly post-treatment were found, and only one was clinically relevant (the increase in nasopharyngeal area) (Appendix A). However, it must be noted that only two small studies were included in this review, which might indicate an absence of evidence, and not necessarily evidence of absence. Chin-cup treatment has been reported to induce certain skeletal adaptations [44,45], the long-term stability of which, however, is questionable. At the present time, there is not enough evidence to suggest that chin-cup treatment negatively influences airway dimensions.

### 4.2. Strengths and Limitations

This systematic review has several strengths, comprising an a priori registered protocol [14], a comprehensive literature search, the inclusion of an untreated Class III control group, the use of modern analytic methods [21], the application of the GRADE approach to assess the strength of provided recommendations [17], and the transparent provision of all data [46].

Some limitations exist nonetheless. For one, methodological issues, which might influence the present conclusions, existed in all the included studies. This is especially the case in the included retrospective nonrandomized studies [47,48]. The inclusion of nonrandomized studies in meta-analyses is not considered prohibitory per se, provided that robust bias appraisal has been performed and recent guidance has been provided on how to appropriately incorporate such designs [24]. Furthermore, most meta-analyses were predominantly based on small trials, which might affect the precision of the estimates [49]. Additionally, the small number of trials ultimately included in the meta-analyses and their incomplete reporting of results and potential confounders (baseline malocclusion severity or other patient characteristics and different orthopedic treatment protocols) prevented us from conducting many analyses for subgroups and meta-regressions. Last but not least, this study, although it was originally otherwise planned, only included lateral cephalograms for evaluating airway dimensions because no eligible controlled study with CBCTs could be found. Lateral cephalograms are abundantly available because they are part of patients’ usual records, but only a moderately high correlation is to be expected between cephalometric and CBCT measurements in the assessment of the airway dimensions [50].

## 5. Conclusions

Current evidence indicates that orthopedic treatment with maxillary protraction for Class III malocclusion might be associated with increased dimensions of the upper airways, which seem to be retained after treatment. However, our confidence in these data is very low due to the poor quality of existing studies and their small number. Restriction of mandibular growth with chin-cup seems likewise to be associated to some extent with increased airway dimensions, but these effects are less pronounced. It is crucial that the clinical relevance of such anatomical changes be confirmed by functional analyses of breathing ability before concrete recommendations can be formulated.

## Figures and Tables

**Figure 1 jcm-09-03015-f001:**
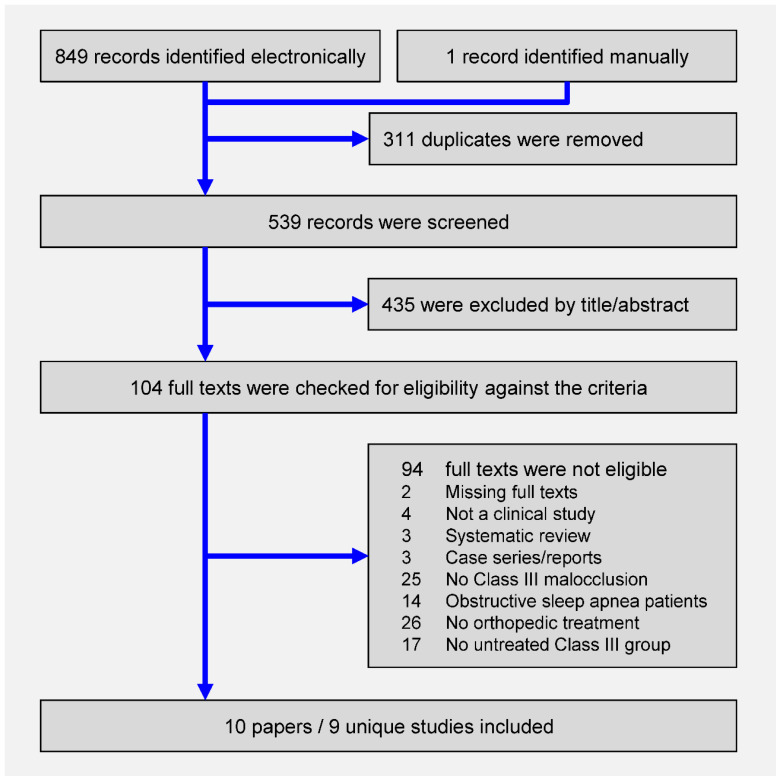
Preferred Reporting Items for Systematic Reviews and Meta-Analyses flow diagram for the identification and selection of studies.

**Table 1 jcm-09-03015-t001:** Characteristics of the included studies.

Study	Design; Setting; Country *	Patients (M/F); Age †	Inclusion Criteria	Appliance (Active Duration)	FU	Imaging Modality
Akin [28]	rNRS; Uni; TR	EG1: 25 (10/15); 10.3 EG2: 25 (9/16); 9.8 CG: 17 (8/9); 10.1	EG1-2/CG: −5° ≤ ANB ≤ 0°; Cl. III MoRel; edge-to-edge or aCB InRel; flat/mildly concave profile; successful Tx	EG1: FM + RME (NR) EG2: CC (NR) CG: (6.0 mos)	-	LC
Tuncer [9]	rNRS; Uni TR	EG: 20 (10/10); 10.3 CG: 18 (10/8); 9.9	EG/CG: mandibular prognathism without shift; neutral SN-ML	EG: CC + BP (9.8 mos) CG: (11.1)	-	LC
Tuncer [29]	rNRS; Uni TR	EG1: 17 (9/8); 11.3 EG2: 17 (10/7); 11.5 CG: 11(8/3); 9.1	EG1-2/CG: ANB < 0°; SNA < 82°; aCB InRel; HWR between PP2 and MP3cap; neutral (EG1)/high SN-ML (EG2);	EG1: FM + BP (9.7 mos) EG2: FM + BP (10.6 mos) CG: (12.0 mos)	-	LC
Danaei [30]	rNRS; Uni; IR	EG: 19 (6/13); 7.9 CG: 15 (4/11); 7.5	EG/CG: SNA < 77°, 76° ≤ SNB ≤ 80°, ANB < 1°; aCB InRel; Cl. III MoRel; no shift	EG: FM (10.5 mos) CG: (10.5 mos)	-	LC
Kilinc [31]	rNRS; Uni TR	EG: 18 (7/11); 10.5 CG: 17 (8/9); 10.9	EG/CG: maxillary skeletal retrusion; aCB InRel; Cl. III MoRel; HWR between PP2 and MP3cap	EG: FM + RME (6.9 mos) CG: (9.8 mos)	-	LC
Lombardo [32]	rNRS; Uni; IT	EG: 47 (22/25); 7.8 CG: 18 (9/9); 8.9	EG/CG: edge-to-edge or aCB InRel; Cl. III MoRel; WITS ≤ −2 mm; no shift; CVM stage 1–3	EG: FM + RME + BP (24.0 mos) CG: (25.2 mos)	5.4 yrs	LC
Menendez-Diaz [33]	rNRS; Uni ES	EG: 64 (10/34); 8.1 CG: 14 (8/6); 8.2	EG/CG: Cl. III; no shift	EG: FM + RME (18.0 mos) CG: (23.2 mos)	-	LC
Mucedero [11]; Baccetti [12]	rNRS; Uni; IT	EG1: 22 (10/12); 8.9 EG2: 17 (10/7); 7.1 CG: 20 (8/12); 8.1	EG1-2/CG: WITS ≤ −2 mm; edge-to-edge or aCB InRel; Cl. III MoRel	EG1: FM + BP (19.2 mos) EG2: FM + RME (25.2 mos) CG: (22.8 mos)	2.1 yrs	LC
Yagci [34]	pNRS; Uni; TR	EG1: 15 (7/8); 9.6 EG2: 15 (8/7); 9.5 CG: 15 (8/7); 9.8	EG/CG: Cl. III MoRel; edge-to-edge or aCB InRel; ANB ≤ 0°; N-Aperp ≤ −2 mm	EG1: FM1 + RME (13.4 mos) EG2: FM2 + RME (14.9 mos) CG: (11.64)	-	LC

* given with the country’s ISO 3166 alpha-2 code. † given as mean (one value) or if mean not reported given as range (two values in parenthesis). aCB, anterior cross bite; ANB, A point-Nasion-B point angle; BP, Bite plane or bite block; CC, chin cup; CG, control group; Cl., Angle’s Class; CVM, cervical vertebrae maturation index; EG, experimental group; FM, facemask or other maxillary protraction appliance; HWR, hand-wrist radiograph; InRel, incisor relationship; LC, lateral ceph; mo, month; MoRel, molar relationship; MP3, middle phalanx of the third finger; NR, not reported; pNRS; prospective nonrandomized study; PP2, proximal phalanx of the second finger; RME, rapid maxillary expansion; rNRS, retrospective nonrandomized study; SNA, Sella-Nasion-A point angle; SNB, Sella-Nasion-B point angle; SN-ML, Sella-Nasion-mandibular plane angle; Tx, treatment; Uni, university clinic; yr, year.

**Table 2 jcm-09-03015-t002:** Detailed assessment of included nonrandomized studies with the ROBINS-I tool.

Reference	Akin [28]	Tuncer [9]	Tuncer [29]	Danaei [30]	Kilinc [31]	Lombardo [32]	Menendez-Diaz [33]	Mucedero [11]; Baccetti [12]	Yagci [34]
Was the study prospective?	N	N	N	N	N	N	N	N	Y
Was selection of patients based on any factor that could influence airways post treatment (age, sex, skeletal configuration, compliance, breakages)?	Y	PY	Y	Y	PY	PN	PY	PY	Y
Were treated/untreated groups clearly defined?	Y	Y	Y	Y	Y	Y	Y	Y	Y
Was the observation period similar for treated/untreated patients?	NI	Y	PN	PY	PN	Y	PN	PN	PY
Were treated/untreated patients similar in terms of baseline age?	Y	PY	PN	Y	Y	PY	Y	PY	Y
Were treated/untreated patients similar in terms of baseline sex?	Y	PY	PN	Y	N	PY	N	N	Y
Were treated/untreated patients similar in terms of dental/skeletal malocclusion?	Y	PY	N	NI	Y	PY	Y	Y	Y
Were treated/untreated patients similar in terms of baseline airways?	PN	PN	Y	Y	Y	PY	Y	Y	N
Was the use of any other appliances/adjuncts the same among treated/untreated patients?	NA	NA	NI	NI	NA	NA	NI	NA	NA
Was outcome measurement similar for treated/untreated patients?	Y	Y	Y	Y	Y	Y	Y	Y	Y
Was outcome measurement done blindly for both treated/untreated patients?	N	N	N	N	N	N	N	N	N
Were treated/untreated patients treated/observed at the same place/time?	NI	NI	N	NI	N	N	N	N	Y

N, no; NA, not applicable; NI, no information; PN, probably not; PY, probably yes; Y, yes.

**Table 3 jcm-09-03015-t003:** Random-effects meta-analyses on the effect of facemask (with/without maxillary expansion) on upper airway dimensions.

Outcome	*n*	Effect (95% CI)	*p*	I^2^ (95% CI)	tau^2^ (95% CI)	95% Prediction
**Post-treatment**						
Total nasopharyngeal area (mm^2^)	4	SMD = 1.62 (1.20, 2.04)	**<0.001**	23% (0%, 91%)	0.04 (0, 1.42)	0.34, 2.90
Adenoidal nasopharyngeal area (mm^2^)	2	MD = 0.34 (−0.10, 0.77)	0.13	0% (0%, 99%)	0 (0, 7.60)	-
Aerial nasopharyngeal area (mm^2^)	2	MD = 1.29 (0.80, 1.77)	**<0.001**	0% (0%, 99%)	0 (0, 14.38)	-
Oropharyngeal area (mm^2^)	2	MD = −0.18 (−1.65, 1.29)	0.81	89% (24%, 100%)	1.00 (0.04, 141.23)	-
Upper adenoid size (AD2-H; mm)	3	MD = 0.59 (−0.52, 1.70)	0.30	57% (0%, 98%)	0.55 (0, 24.22)	−11.24, 12.42
Lower adenoid size (AD1-Ba; mm)	3	MD = 0.12 (−2.20, 2.44)	0.92	76% (0%, 99%)	3.11 (0, 76.33)	−26.88, 27.12
Upper airway dimension (PNS-AD2; mm)	6	MD = 2.45 (0.97, 3.92)	**0.001**	87% (64%, 97%)	2.83 (0.74, 15.17)	−2.67, 7.57
Lower airway dimension (PNS-AD1; mm)	6	MD = 2.10 (1.50, 2.70)	**<0.001**	5% (0%, 87%)	0.04 (0, 4.83)	1.11, 3.09
McNamara’s upper pharynx dimension (mm)	6	MD = 1.59 (0.57, 2.62)	**0.002**	73% (15%, 95%)	1.08 (0.07, 7.56)	−1.63, 4.82
McNamara’s lower pharynx dimension (mm)	6	MD = 1.02 (0.17, 1.88)	**0.02**	70% (16%, 94%)	0.69 (0.06, 4.90)	−1.58, 3.63
**Postretention**						
Upper adenoid size (AD2-H; mm)	2	MD = −1.13 (−4.25, 2.00)	0.48	72% (0%, 100%)	3.70 (0, 641.91)	-
Lower adenoid size (AD1-Ba; mm)	2	MD = −2.67 (−4.63, −0.70)	**0.008**	14% (0%, 99%)	0.31 (0, 275.16)	-
Upper airway dimension (PNS-AD2; mm)	2	MD = 3.71 (0.80, 6.62)	**0.01**	65% (0%, 100%)	2.91 (0, 563.83)	-
Lower airway dimension (PNS-AD1; mm)	2	MD = 3.59 (1.75, 5.44)	**<0.001**	0% (0%, 98%)	0 (0, 132.16)	-
McNamara’s upper pharynx dimension (mm)	2	MD = 2.27 (0.80, 3.74)	**0.003**	0% (0%, 98%)	0 (0, 89.15)	-
McNamara’s lower pharynx dimension (mm)	2	MD = 1.84 (−2.08, 5.75)	0.36	85% (NC)	6.80 (NC)	-

CI, confidence interval; MD, mean difference; SMD, standardized mean difference. Statistical significance is denoted in bold.

**Table 4 jcm-09-03015-t004:** *p* values from subgroup analyses and meta-regressions.

Outcome	Age	Male%	Baseline Airway	RME	Tx Duration
Upper airway dimension (PNS-AD2; mm)	0.45	0.65	0.69	0.29	0.27
Lower airway dimension (PNS-AD1; mm)	0.98	0.15	0.37	0.65	0.26
McNamara’s upper pharynx dimension (mm)	0.46	0.23	0.46	0.81	0.33
McNamara’s lower pharynx dimension (mm)	0.13	0.65	0.45	0.70	0.67

RME, rapid maxillary expansion, Tx, treatment.

**Table 5 jcm-09-03015-t005:** Summary of Findings Table according to GRADE approach.

	Anticipated Absolute Effects (95% CI)			
**Outcome** **Studies (Patients)**	**Control (Growth)**	**Maxillary Protraction**	**Quality of the Evidence (GRADE) ^b^**	**What Happens with Maxillary Protraction**
Total airway area 35 patients (1 study)	−38.4 mm ^2^	223 mm^2^ greater (14.0 to 431.7 mm^2^ greater)	○○○○ very low ^c,d^ due to bias, imprecision	Might be associated with greater airway area
Upper airway dimensions 316 patients (6 studies)	0.3 mm ^a^	2.5 mm greater (1.0 to 3.9 mm greater)	○○○○ very low ^c^ due to bias	Might be associated with greater upper airway dimensions
Lower airway dimensions 316 patients (6 studies)	0.5 mm ^a^	2.1 mm greater (1.5 to 2.7 mm greater)	○○○○ very low ^c^ due to bias	Might be associated with greater lower airway dimensions
Upper pharynx dimensions (McNamara’s) 323 patients (6 studies)	0.6 mm ^a^	1.6 mm greater (0.6 to 2.6 mm greater)	○○○○ very low ^c^ due to bias	Might be associated with greater upper pharynx dimensions
Lower pharynx dimensions (McNamara’s) 323 patients (6 studies)	0.1 mm ^a^	1.0 mm greater (0.2 to 13.9 mm greater)	○○○○ very low ^c^ due to bias	Might be associated with greater lower pharynx dimensions

Intervention: orthopedic maxillary protraction with facemask or reverse headgear/Population: pre-adolescent children with skeletal Class III malocclusion/Setting: university clinics (Iran, Italy, Spain, Turkey). ^a^ Response in the control group is based on random-effects meta-analysis of the control groups of included studies. ^b^ Starts from “high”. ^c^ Downgraded by three levels for bias due to the inclusion of retrospective nonrandomized studies with serious risk of bias. ^d^ Downgraded by one level for imprecision due to the inclusion of an inadequate sample. CI, confidence interval; GRADE, Grading of Recommendations Assessment, Development and Evaluation, ○○○○, downgraded by 4 points.

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
