# Peer review of "Effect of Orthopedic Treatment for Class III Malocclusion on Upper Airways: A Systematic Review and Meta-Analysis"

_jcm, 2020, doi:10.3390/jcm9093015_

Round 1
Reviewer 1 Report
Why was country considered in study characteristics?
It seems that the finally selected papers are relatively recent papers. As far as I know, CBCT is also taken routinely as the lateral cephalogram, and CBCT and various 3D programs have been actively progressed for analyzing changes in airway. Why did you select articles using the lateral cephalogram? Was there any specific reason other than that the lateral cephalogram is in general acquirement?
For each paper, was there any agreement on the location of determining the compartment of the airway or the location of measurement? The expression "dimension" on this paper seems a bit vague.
The facemask effect has the protraction effect of the maxilla, but there is also a clockwise rotation of mandible. In this paper, however, only maxillary protraction is mentioned as the effect of facemask. The chin-cup causes the mandible to rotate backward and downward and has the effect of restriction of mandibular growth. You concluded that chin-cup had a smaller benefit on the airway dimension in this paper. My question is, does the author think that the rotation of the mandible does not have a significant effect on the change in the airway dimension? What do you think the increase of airway dimension is not by facemask only, but by the contribution of maxillary expansion?
I agree with the phrase "clinical relevance of such anatomical changes is confirmed by functional analyses of breathing" in your conclusion. Why did you exclude the 14 papers about OSA, while mentioning the correlation of airway dimensional change and breathing?
Reviewer 2 Report
The authors, through a systematic review, made a reasonable attempt to accurately assess the effect of different orthopedic appliances on the upper airway dimension in growing patients with class III malocclusion.
I congratulate the authors for this very relevant research, which will add to the orthodontics field.
The study is of sound design and of clear practical and clinical interest, but some improvements are needed.
INTRODUCTION
The introduction section is well written and provide sufficient background. However, adequate references to related and previous work still missing.
I suggest to add the following 3 references in the manuscript about the evaluation of the upper airways by means the use of CBCT and the efficiency of RME-face mask in skeletal Class III patients.
https://doi.org/10.3390/ma13102239
https://doi.org/10.3390/ma13041007
PMID: 22971261
RESULTS line 163: As mentioned by the Auhtors, all studies included in this review assessed the effect of treatment on the upper airways with lateral cephalograms. I am surprised that no Author in the literature has investigated three-dimensional changes produced by orthopedic appliances on the upper airways yet. I found this article that should be included in your review.
https://doi.org/10.1371/journal.pone.0135273
DISCUSSION
line 297: Unfortunately I cannot agree with them, as some studies performed on CBCT exist in the literature and should be included in the current study. The three-dimensional measurements performed on CBCT could be extremely different. In fact, different authors stated that the evaluation of the upper airways performed on lateral cephalograms is less precise than those performed on three-dimensional imaging methods.
I suggest to the authors to include in the limitations of the study that most of the studies considered are performed on two-dimensional radiographs and therefore not very reliable. Furthermore, I suggest that the authors carefully search the literature for other articles performed on three-dimensional imaging technique. In the event that the research is limited to a few articles, I suggest the Authors to highlight in the limitations section that currently few three-dimensional studies have been conducted and therefore the conclusions of the review may be inaccurate. It is expected in the future to have available a greater number of studies performed on three-dimensional methods so as to be able to perform new systematic reviews.
I also suggest to the Authors to add the concept that the reduced number of articles could be due to the biological cost of the CBCT exam, the development of three-dimensional radiation-free methods will certainly favor in the future to conduct a greater number of studies. I suggest to the Authors to cite the following article:
https://doi.org/10.1186/s40510-019-0293-x
Round 2
Reviewer 1 Report
It is judged to have been appropriately modified according to the reviewer's request. It seems to be a clinically useful study.
Reviewer 2 Report
Dear Authors
The article has been improved.
Kind regards